# The Effects of Stress on Hippocampal Neurogenesis and Behavior in the Absence of Lipocalin-2

**DOI:** 10.3390/ijms242115537

**Published:** 2023-10-24

**Authors:** Ana Catarina Ferreira, Fernanda Marques

**Affiliations:** 1Life and Health Sciences Research Institute (ICVS), School of Medicine, University of Minho, Campus Gualtar, 4710-057 Braga, Portugal; catarina.ferreira.mail@gmail.com; 2ICVS/3B’s—PT Government Associate Laboratory, 4710-057 Braga, Portugal

**Keywords:** lipocalin-2, hippocampal neurogenesis, corticosterone, anxiety, memory

## Abstract

Lipocalin-2 (LCN2) is an acute phase protein able to bind iron when complexed with bacterial siderophores. The recent identification of a mammalian siderophore also suggested a physiological role for LCN2 in the regulation of iron levels and redox state. In the central nervous system, the deletion of LCN2 induces deficits in neural stem cells proliferation and commitment, with an impact on the hippocampal-dependent contextual fear discriminative task. Additionally, stress is a well-known regulator of cell genesis and is known to decrease adult hippocampal cell proliferation and neurogenesis. Although voluntary running, another well-known regulator of neurogenesis, is sufficient to rescue the defective hippocampal neurogenesis and behavior in LCN2-null mice by promoting stem cells’ cell cycle progression and maturation, the relevance of LCN2-regulated hippocampal neurogenesis in response to stress has never been explored. Here, we show a lack of response by LCN2-null mice to the effects of chronic stress exposure at the cellular and behavioral levels. Together, these findings implicate LCN2 as a relevant mediator of neuronal plasticity and brain function in the adult mammalian brain.

## 1. Introduction

In the adult mammalian brain, new neurons are continuously generated from resident neural progenitors at the subgranular zone (SGZ) of the dentate gyrus (DG) in the hippocampus. In rodents, these newly generated functional neurons integrate into the pre-existing neuronal circuitry and contribute to local neural plasticity [1]. The generation of new neurons and their integration is crucial for hippocampal integrity and function, and it has been shown to directly modulate learning and memory, pattern separation, emotion, and even neurodegeneration [2,3,4,5].

Several factors have been described to regulate adult hippocampal neurogenesis and associated behaviors. Among them, lipocalin-2 (LCN2) has been shown to modulate several aspects of the central nervous system, including neurogenesis. As we previously described [6], under physiological conditions, LCN2-null mice display increased anxiety and depressive-like behaviors, with concomitant neuronal structural changes and decreased hippocampal long-term potentiation within the hippocampus [6]. Whether the LCN2 effects are mediated through the modulation of the level of corticosteroids that we found slightly increased in LCN2-null animals [6] or through a novel mechanism still deserves further analysis. More recently, LCN2 was shown to be essential for the redox control of adult hippocampal neural stem cells (NSCs), with an impact on cell cycle progression and death, self-renewal, and differentiation, resulting in altered contextual discriminative behavior [7].

Stress and its end-effectors, glucocorticoids (i.e., corticosterone, CORT), have been shown to negatively impact hippocampal function and structure [8], and, particularly in the process of neurogenesis, it was described to suppress cell proliferation [9,10] and decrease the survival of newborn cells in the DG [11], culminating in the precipitation of anxiety- and depressive-like behaviors [10]. Of interest, LCN2 has been said to have roles in the mediation of stress-induced responses at the cellular and behavioral levels [12,13], but the contribution of hippocampal neurogenesis in the control of such responses by LCN2 is not known.

Considering the importance of LCN2 in critical steps of NSC physiology, adult hippocampal neurogenesis, and animal behavior, we evaluated the effects of stress in the modulation of hippocampal neurogenesis and behavior in the absence of LCN2. For that, we used chronic CORT administration to decrease hippocampal neurogenesis in LCN2-null mice. Here, we show that in LCN2-null mice, neurogenesis and behavior were not affected the exposure to chronic CORT.

## 2. Results

### 2.1. Effectiveness of Chronic CORT Administration

In order to decrease hippocampal neurogenesis and analyze its cellular and behavioral effects in the absence of LCN2, we exposed wild-type (Wt) and LCN2-null mice to a chronic treatment of CORT (20 mg/kg) for 28 days (Figure 1a) and further analyzed the biological, morphological, and behavioral signatures of the stress response.

Firstly, we assessed the biological efficacy of the chronic CORT treatment. For that, we monitored body weight gain once a week during the 4 weeks of CORT injections and the thymus and adrenals weight at the sacrifice. We observed that the CORT exposure significantly affected the percentage of body weight gain over the weeks of treatment (CORT effect: F_3,135_ = 10.4, *p* < 0.0001), with a tendency to a more pronounced effect in CORT Wt animals (Figure 1b). Similarly, the CORT treatment altered adrenal (F_1,17_ = 81.4, *p* < 0.0001) and thymus (F_1,17_ = 76.9, *p* < 0.0001) weight (Figure 1c,d), irrespectively of the animals’ genotype (adrenals weight*genotype: F_1,17_ = 1.14, *p* = 0.30; thymus weight*genotype: F_1,17_ = 3.09, *p* = 0.09).

### 2.2. Chronic CORT Administration Impairs Cell Proliferation and Reduces the Pool of Stem Cells

Consistent with the current knowledge on the negative effects of chronic stress and its end-effectors, glucocorticoids, on adult hippocampal neurogenesis [14], we next examined the impact of chronic CORT exposure on cell proliferation and survival in the absence of LCN2. We observed that chronic CORT administration impacted the number of proliferating Ki67^+^ cells in the DG (CORT effect: F_1,14_ = 3.45, *p* = 0.08) by specifically decreasing cell proliferation in the Wt animals (*p* = 0.0002 versus vehicle Wt; Figure 2a,b). Additionally, an analysis of cell survival, as the number of BrdU^+^ cells in the subgranular zone (SGZ), revealed that the CORT treatment had no significant impact on this parameter (CORT effect: F_1,14_ = 0.31, *p* = 0.58). Of notice, both the cell proliferation and survival analyses in the LCN2-null mice SGZ after CORT treatment revealed no significant effects (Figure 2a,b). We observed a decrease in Ki67^+^ and BrdU^+^ cells in vehicle LCN2-null mice, as previously reported [7], but both populations were not affected by the CORT treatment.

We next analyzed the effect of chronic CORT exposure on the pool of stem cells. When labelling type-1 radial glia-like stem cells, by co-staining with BrdU and glial fibrillary acidic protein (GFAP), we observed that CORT induced a significant depletion on the number of type-1 stem cells (CORT effect: F_1,10_ = 22.54, *p* = 0.0008), regardless of the animals’ genotype (CORT*genotype: F_1,10_ = 27.2, *p* = 0.0004). Specifically, CORT induced a significant decrease in the number of type-1 GFAP^+^ BrdU^+^ stem cells in both the Wt (*p* = 0.01 versus vehicle Wt) and LCN2-null mice (*p* < 0.0001 versus vehicle LCN2-null; Figure 2c,d). Importantly, this was consistent with the analysis of nonradial Sox2^+^ Ki67^-^ type-1 stem cells (CORT effect: F_1,15_ = 10.23, *p* = 0.006; *p* = 0.05 Wt vehicle versus CORT, *p* = 0.02 LCN2-null vehicle versus CORT; Appendix A). On the other hand, an analysis of type-2 stem cells revealed no major overall effect of stress exposure (CORT effect: F_1,15_ = 0.90, *p* = 0.36) but a significant reduction in the number of Sox2^+^ Ki67^+^ proliferating progenitor cells in the Wt SGZ (*p* = 0.007 versus vehicle Wt; Figure 2c,d). In LCN2-null mice, this specific population remained similar to the vehicle group (Figure 2c,d).

### 2.3. LCN2 Is Required for the Deleterious Effects of CORT on Behavior and Hippocampal Neurogenesis

To disclose the importance of LCN2 in CORT-induced behavioral deficits, as others have described [15], we performed a behavior analysis after 28 days of CORT administration, which included an assessment of anxiety-like behaviors and of contextual discrimination. Specifically for anxiety, and in the elevated plus maze (EPM) paradigm, chronic CORT exposure induced an anxious-like behavior in Wt mice (*p* = 0.004 versus vehicle Wt), as observed by the decreased percentage of time spent in the open arms, whereas LCN2-null mice behaved as in the vehicle conditions (Figure 3a). In addition, the CORT treatment did not affect general locomotor activity in all groups (Figure 3a). Accordingly, when animals were assessed in the novelty-suppressed feeding (NSF) test, the CORT Wt animals significantly increased their latency of time to feed (*p* = 0.04 versus vehicle Wt), indicating an anxious-like behavior induced by CORT exposure (Figure 3b), which again was not observed in the CORT LCN2-null mice (Figure 3b). No major effects on overall appetite drive were observed (Figure 3b).

Moreover, an analysis of the animals’ capacity to discriminate between contexts in the CFC test showed that the CORT administration influenced contextual retrieval (CORT effect: F_1,17_ = 3.3, *p* = 0.08; Figure 3c) and discrimination (CORT effect: F_1,15_ = 5.9, *p* = 0.03; Figure 3d). Particularly, CORT led to a significant decrease in the percentage of freezing upon re-exposure to training context A in Wt animals (*p* = 0.03 versus vehicle Wt; Figure 3c), also decreasing their ability to discriminate between the different contexts (*p* = 0.02; Figure 3d). Again, CORT had no significant effect on LCN2-null mice (Figure 3c,d).

In addition, we observed that impaired neurogenesis induced by CORT requires LCN2 for behavioral outcomes. Specifically, chronic CORT treatment negatively affected the generation of newborn neurons (CORT effect: F_1,17_ = 5.2, *p* = 0.04). The quantification of the total number of Calb^+^ BrdU^+^ cells showed that CORT promoted a significant decrease in the number of new cells in the Wt mice (58% less, *p* = 0.04 versus vehicle Wt), whereas it remained unchanged in the LCN2-null mice (Figure 3e,f). Together, our data reveal that an LCN2 absence blocks glucocorticoids-driven anxiety and cognitive function, as it prevents CORT-promoted hippocampal neurogenesis decline.

## 3. Discussion

A lack of LCN2 was recently described to induce deficits in NSC proliferation and commitment by modulating the levels of oxidative stress, with an impact on hippocampal-dependent contextual fear discrimination [7] and anxiety [6]. Of interest, those effects were rescued by voluntary running, which is a well-known regulator of cell genesis, by counteracting oxidative stress in NSCs, which was detected in LCN2-null animals (Ferreira et al. 2019). Still, the effect of hippocampal neurogenesis on external regulation by stress in the absence of LCN2 is lacking. Here, we show evidence for the effects of neurogenesis modulation through the chronic administration of CORT in a mouse model of LCN2-null-impaired cell genesis. Specifically, we found a lack of response by LCN2-null mice to chronic CORT administration. We show that CORT differentially affected cell proliferation and survival in Wt and LCN2-null mice, and glucocorticoids-driven anxiety and poor contextual discriminative behaviors were inexistent in the absence of LCN2.

Stress triggers a variety of adaptive cellular responses that help to control brain homeostasis and shape the adequate behavioral response of an animal, and failure to properly adjust such responses often results in affective disorders. In this context, accumulating evidence also suggests that chronic stress exposure negatively affects hippocampal neurogenesis [16,17] and highly contributes to anxiety-like behaviors [15,18,19] and memory impairment [20]. Noticeably, Lcn2 has been shown to be up-regulated in the hippocampus after restraint stress to control neuronal excitability and anxiety behavior [12], and, in physiological conditions, LCN2-null mice present mood and cognitive alterations [6] and impaired cell genesis [7]. Taking this into consideration, we evaluated the impact of stress-related hormones in the adult hippocampal neurogenesis of LCN2-null mice. For that, we injected Wt and LCN2-null mice with CORT daily for 28 days, which is an alternative to the use of chronic stress paradigms, which has been shown to be effective in inducing altered mood behaviors [15,18], decreasing hippocampal neurogenesis and body weight, and maintaining high levels of CORT [17,21]. The dosage and regiment of the CORT injections were performed as previously reported [22] and, in fact, we confirmed the efficacy of the stress protocol applied here since both Wt and LCN2-null mice presented significant alterations in body, adrenal, and thymus weight. At the cellular level, we observed that CORT negatively impacted the NSC population, as it induced a significant depletion of type-1 and -2 stem cells on both Wt and LCN2-null mice. Furthermore, it affected the generation of new neurons, but only in Wt mice, which translated into increased anxiety and impaired contextual discrimination. Of notice, this was not observed in the LCN2-null mice after treatment with CORT. However, this is well described, and as we confirm, this is an effective model to induce stress responses at the behavioral and cellular levels. However, other models should be considered, such as the use of the chronic, unpredictable stress protocol or even the osmotic pumps to replace the injections that these animals received.

Glucocorticoids promote their regulation on both brain and peripheral functions primarily via their two specific receptors, the glucocorticoid and the mineralocorticoid receptors. Noticeably, these receptors are highly expressed by neural precursor cells, particularly by quiescent type-1 NSCs and type-2a amplifying progenitors, which renders them more susceptible to the levels of circulating glucocorticoids [23]. It is, therefore, not surprising to observe a reduction in the pool of stem cells upon CORT injection, which is consistent with other reports [24]. Still, at this point, it is not clear what the fate of these cells is. Of interest, stress and its related glucocorticoid hormones are known to impact cell proliferation and neurogenesis either by potentiating apoptotic cell death [24] or by promoting the cell cycle arrest of progenitor cells [25], which should be further clarified in our model of stress.

The observations that CORT injections had no effect on hippocampal neurogenesis and behavior in the absence of LCN2 are not surprising if we consider the fact that LCN2-null mice, in physiological conditions, show increased CORT levels [6]. We have previously described that LCN2-null mice present an overactivation of the hypothalamic–pituitary–adrenal axis [6], translated into a sightly sustained production of CORT, which might explain their incapacity to respond to CORT. Our previous description that LCN2-null mice, in basal states, present anxious and depressive-like behaviors, impaired cognition, and long-term potentiation, as well as altered hippocampal cytoarchitecture [6], largely recapitulate some of the features observed in animals after high glucocorticoid/stress exposure [26,27,28]. As such, we cannot exclude the contribution of the sustained production of CORT in the absence of LCN2 in boosting hippocampal neurogenesis deficits and in promoting behavioral deficits that these animals presented already in basal conditions [7]. Certainly, we must consider that the effects of stress largely depend on the stress duration and paradigm used, as others have reported LCN2-null mice to present enhanced stress-induced anxiety after restraint stress [12]. Nevertheless, we show here that LCN2 has important roles in mediating the deleterious effects of prolonged chronic exposure to glucocorticoids on neurogenesis and behavior.

## 4. Materials and Methods

### 4.1. Animal Experiments

Experiments were conducted on 2-month-old male mice lacking LCN2 (LCN2-null) and their respective wild-type (Wt) littermate controls in a C57BL/6J mouse background. Animals were obtained from crossing heterozygous animals, and mice’s genotype was confirmed by polymerase chain reaction. The use of only males of 2 months of age is mainly related to the fact that, in previous studies, we have described that LCN2-null males of this age presented altered hippocampal neurogenesis and contextual discriminative behaviors [7]; however, it is not expected that this is gender- or age-specific since the cognitive deficits and hippocampal neurogenesis defects were also detected in LCN2-null aged animals [29]. Mice were housed and maintained according to the guidelines for the care and handling of laboratory animals in Directive 2010/63/EU of the European Parliament and of the Council, in a controlled environment at 22–24 °C and 55% humidity, on 12 h light/dark cycles, and fed with regular rodent’s chow and tap water ad libitum. All animal procedures were conducted in accordance with the Portuguese national authority for animal experimentation, Direção Geral de Alimentação e Veterinária (ID: DGAV9457).

Regarding the number of animals used in this study, for the behavioral analysis, 6–10 mice per group were used, and for the assessments of cell proliferation and neural stem cell self-renewal and survival, 4–6 mice per group were used.

### 4.2. Corticosterone Injections

The repeated injection of the major stress hormone CORT in rodents provides a reliable animal model for studying stress-triggered responses [30,31]. Thus, to analyze the effect of chronic exposure to glucocorticoids on mediating cell survival and behavior in the absence of LCN2, both Wt and LCN2-null mice were subcutaneously injected with CORT (20 mg/kg, dissolved in sesame oil; Sigma Aldrich, St. Louis, MO, USA) daily for 28 days (CORT group) (Figure 1a; [22,32]). An additional group of animals of both genotypes were subcutaneously injected with the vehicle (sesame oil; vehicle group) for the same period. As described above, experimental groups were intraperitoneally injected with 50 mg/kg of BrdU (Sigma Aldrich), previously dissolved in saline solution, twice a day for 5 consecutive days at the beginning of the CORT treatment to assess neuron survival and maturation. Animals were weekly monitored for body weight gain during the protocol, and adrenal and thymus weight were measured at the time of sacrifice. The behavioral analysis occurred at the end of the protocol.

### 4.3. Behavior

#### 4.3.1. Novelty-Suppressed Feeding

To examine anxiety-like behavior in a novel environment, animals were assessed in the novelty-suppressed feeding (NSF) paradigm. After 18 h of food deprivation, animals were placed in an open field arena (Med Associates Inc., St. Albans, VT, USA), where a single food pellet was placed in the center. The latency of time to feed was recorded and used as a measure of anxiety-like behavior. After reaching the pellet, animals returned to the home cage, where they were allowed to eat pre-weighted food for 5, 15, and 30 min as a measure of appetite drive.

#### 4.3.2. Elevated plus Maze

Anxious behavior was analyzed through the elevated plus maze (EPM) test. The behavioral apparatus (ENV-560; Med Associates Inc.) consisted of two opposite open arms (50.8 cm × 10.2 cm) and two closed arms (50.8 cm × 10.2 cm × 40.6 cm) elevated 72.4 cm above the floor and dimly illuminated. Mice were individually placed in the center of the maze and allowed to freely explore it for 5 min. The percentage of time spent in the open arms, monitored through an infrared photobeam system (MedPCIV, Med Associates Inc.), was used as an index of anxiety-like behavior, and the total number of entries in the arms maze was used as an index of general locomotor activity.

#### 4.3.3. Contextual Fear Conditioning

Contextual fear conditioning (CFC) was conducted in ventilated sound-attenuated chambers with internal dimensions of 20 cm wide × 16 cm deep × 20.5 cm high (SR-LAB, San Diego Instruments, San Diego, CA, USA) and a light mounted directly above the chamber to provide illumination. The floor consisted of a stainless grid that was attached to a shock generator (Coulbourn Instruments, Allentown, PA, USA) for foot shock delivery. A fan mounted on one side of each box provided ventilation and background noise, which was only put on upon context changing. Mice behavior was recorded by digital video cameras mounted above the conditioning chamber, and freezing behavior was manually scored by a blind observer using the Etholog V2.2 software [33]. Freezing was defined as the complete absence of motion for a minimum of 1 s.

The fear conditioning procedure was conducted for 2 days. On day 1, mice were placed in the conditioning chamber and received pairings of light and a cotermination shock (1 s, 0.5 mA), spared from each other with an interval of 20 s. Mice received three light–shock pairings with an intertrial interval of 2 min between each block, and the first light presentation started 3 min after the mouse was placed into the chamber. Animals returned to their home cage 30 s after the last shock was presented. The chambers were cleaned with 10% ethanol between each mouse. On the following day, mice were tested for conditioned fear to the training context and placed in the same chamber (context A), where the training contextual conditions remained from the day before, but no presentation of the conditioned stimulus occurred. Mice were placed into the chambers for 3 min, and the entire session was scored for freezing. Two hours after this, animals were presented with a novel context (context B), where the grid was removed and black plastic inserts covered the floor and walls of the chamber. In addition, the chamber was scented with a paper towel dabbed with vanilla extract and placed underneath the chamber floor, and the ventilation fan was set on. In addition, the experimenter wore a different style of gloves and changed their lab coat. Chambers were cleaned with 10% ethanol between runs, and mice were kept in a different holding room before testing. Each mouse was placed into the chamber for 3 min, and freezing was scored for the entire session. Parameters analyzed included the percentage of time freezing during the training session, the total percentage of time freezing in the contexts (A) and (B), and the index of discrimination between contexts as the ratio of percentage of time freezing (contexts A − B)/percentage of time freezing (contexts A + B).

### 4.4. Tissue Preparation and Immunohistochemistry

At the end of the behavioral assessment, mice were anesthetized with an intraperitoneal injection of a mixture of ketamine hydrochloride (150 mg/kg, Imalgene 1000, Boehringer Ingelheim International GmbH, Ingelheim am Rhein, Germany) plus medetomidine (0.3 mg/kg, Dorben vet, Zoetis, Lisboa, Portugal) and transcardially perfused with 0.9% saline, followed by perfusion with cold 4% paraformaldehyde (PFA) solution. Brains were removed, postfixed for 1 h in 4% PFA, cryoprotected in 30% sucrose overnight, and then embedded in an optimal cutting temperature compound (ThermoFisher Scientific, Waltham, MA, USA), snap-frozen, and kept frozen at −20 °C until further sectioning. Posterior serial coronal sections (20 μm) were cut in a cryostat and collected in slides for immunohistochemistry. For BrdU immunostaining, antigen retrieval by heat with 10 mM citrate buffer (Sigma Aldrich) was performed on the fixed tissue sections, followed by DNA denaturation with HCl (Sigma) for 30 min at room temperature (RT). An additional blocking step with 10% normal fetal bovine serum (FBS; Invitrogen, Carlsbad, CA, USA) in a solution of PBS 0.3% Triton X-100 (PBS-T; Sigma) was also performed for 30 min at RT. Primary antibody incubation, diluted in blocking solution, occurred overnight at RT as follows: rabbit anti-glial fibrillary acidic protein (GFAP, 1:200; DAKO, Glostrup, Denmark), rabbit anti-calbindin (1:300; Abcam, Cambridge, UK), rat anti-BrdU (1:100; Abcam), mouse anti-Sox2 (1:200; Abcam), rabbit anti-Ki67 (1:300; Millipore, Billerica, MA, USA), and rabbit anti-glutathione peroxidase 4 (Gpx4, 1:50; Abcam). Fluorescent secondary antibodies (Invitrogen) anti-rat, anti-rabbit, and anti-mouse, combined to Alexa 594 or to Alexa 488, were used to detect the respective primary antibodies at a dilution of 1:500 (in PBS-T) for 2 h at RT. To stain the nucleus, sections were then incubated with 4′,6-diamidino-2-phenylindole (DAPI, 1:1000; Sigma), after which slides were mounted with Immu-Mount (ThermoFisher Scientific).

### 4.5. Confocal Imaging and Quantitative Analysis

Fluorescence images of the DG were acquired using the Olympus FV1000 confocal microscope (Olympus, Hamburg, Germany). Fields were acquired using Z-scan with a step of 1 μm between each confocal plane. All sections prepared for comparison were analyzed at the same time, using the same acquisition parameters. The quantification rates of the parameters analyzed were estimated by the analysis of 3–4 sections per animal. The number of double positive cells was calculated using the Olympus Fluoview FV1000 software (Olympus) and normalized for the respective area (mm^2^).

### 4.6. Statistical Analysis

All experiments were performed and analyzed by the same experimenter, blind to the animals’ genotype or group treatment under assessment. Variables followed a Gaussian distribution, as revealed by the D’Agostino–Pearson normality test. Data are reported as mean ± standard error (S.E.M.). The number of biological replicates (n) is specified in the legend of each figure. Statistically significant differences between groups were determined using a two-way ANOVA, followed by Bonferroni’s multiple comparison test. Values were considered statistically significant for *p* ≤ 0.05 (*, # or δ), *p* ≤ 0.01 (**, ## or δδ), *p* ≤ 0.001 (***, ### or δδδ), and *p* ≤ 0.0001 (****, #### or δδδδ).

## Figures and Tables

**Figure 1 ijms-24-15537-f001:**
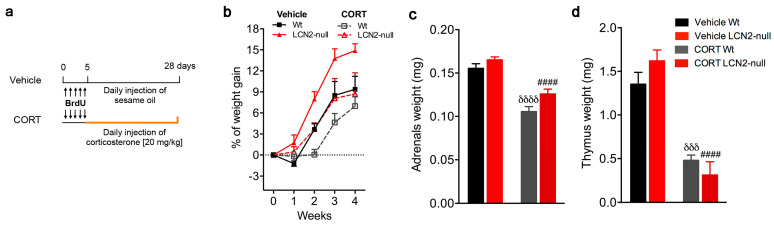
The effects of chronic CORT treatment on biological parameters. (**a**) Representative scheme of the experimental procedure of CORT injections performed. The thymidine analog 5′-bromo-2′-deoxyuridine (BrdU) was injected for 5 days at the beginning of the protocol and, for an additional 28 days, Wt and LCN2-null mice were daily injected subcutaneously with corticosterone (CORT group) or sesame oil (Vehicle group). (**b**) Body weight gain in Wt and LCN2-null mice after chronic CORT injection (n = 6–10 mice per group) were monitored weekly during the protocol of CORT treatment. (**c**,**d**) CORT treatment significantly decreased adrenal and thymus weight, regardless of the genotype. Data are presented as mean ± SEM and were analyzed by two-way ANOVA with Bonferroni’s multiple comparison test. ^δ^ Denotes differences between vehicle and CORT Wt, and ^#^ between vehicle and CORT LCN2-null mice. ^δδδ^
*p* ≤ 0.001, ^####,δδδδ^
*p* ≤ 0.0001.

**Figure 2 ijms-24-15537-f002:**
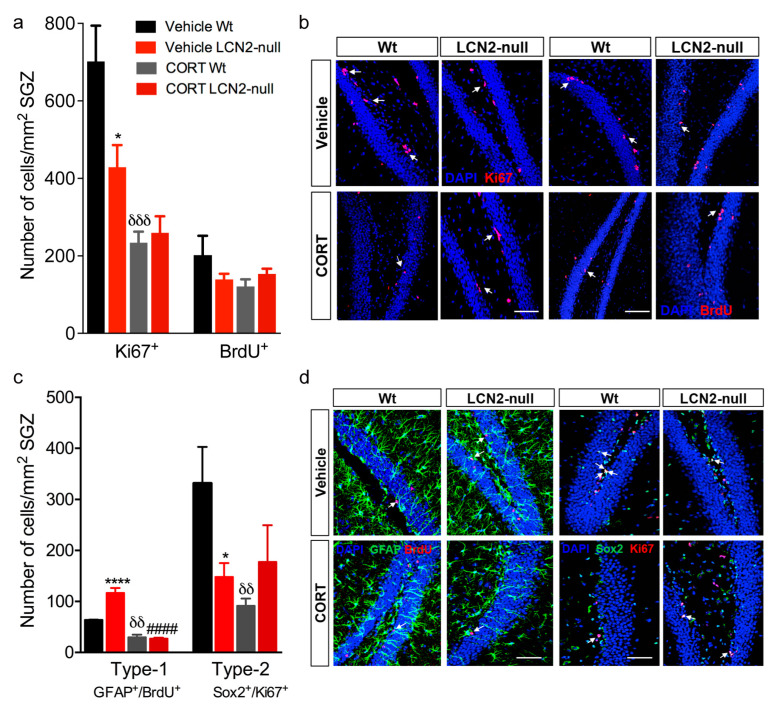
Chronic CORT administration reduces cell proliferation and neural stem cells self-renewal and survival. (**a**) CORT exposure robustly decreased the number of Ki67^+^ proliferative cells, specifically in Wt animals, whereas it did not significantly impact cell survival (BrdU^+^ cells) (n = 4–6 per group). (**b**) Representative images of Ki67 and BrdU immunostaining (indicated by white arrows) in the DG of Wt and LCN2-null mice injected with vehicle and CORT. (**c**) Quantification of the number of radial glia-like type-1 stem cells, as GFAP^+^ BrdU^+^ cells in the SGZ, revealed that CORT treatment induced a significant decrease in both Wt and LCN2-null mice (n = 4 mice per group). The effect of CORT on type-2 stem cells was only evident in Wt animals (n = 5 mice per group). (**d**) Representative images of GFAP^+^/BrdU^+^ and Sox2^+^/Ki67^+^ immunostaining (indicated by white arrows) in the DG of Wt and LCN2-null control mice and those injected with CORT. Scale bars, 100 μm. Data are presented as mean ± SEM and were analyzed by two-way ANOVA with Bonferroni’s multiple comparison test. * Denotes differences between vehicle Wt and LCN2-null mice; ^δ^ between vehicle and CORT Wt; and ^#^ between vehicle and CORT LCN2-null mice. * *p* ≤ 0.05, ^δδ^
*p* ≤ 0.01, ^δδδ^
*p* ≤ 0.001,****^, ####^
*p* ≤ 0.0001.

**Figure 3 ijms-24-15537-f003:**
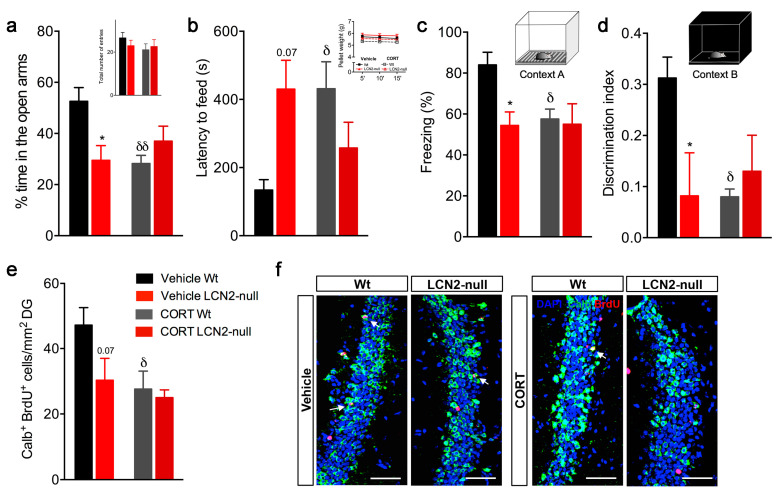
Animal behavior and hippocampal neurogenesis are differentially affected by CORT in Wt and LCN2-null animals. (**a**) Anxiety assessment in the EPM test, after CORT treatment, showed that Wt mice, but not LCN2-null animals, spent less time in the open arms, with no effect among groups on general motor activity (n = 6–10 mice per group). (**b**) In the NSF paradigm, Wt animals increased their latency to feed after the chronic treatment with CORT. No differences were observed between groups in appetite drive (n = 6–10 mice per group). (**c**) Contextual retrieval in context A by Wt mice was impaired after CORT treatment (n = 6–10 mice per group) but not by LCN2-null mice. (**d**) Discrimination index was affected by the chronic CORT treatment, but only in Wt animals (n = 6–10 mice per group). (**e**) Quantitative analysis of the number of newborn mature neurons in DG, as Calb^+^ BrdU^+^ cells, revealed that CORT treatment induced a significant decrease in Wt mice (n = 5 mice per group), with no effect on LCN2-null DG. (**f**) Representative images of calbindin and BrdU immunostaining (indicated by white arrows) in DG of Wt and LCN2-null vehicle mice and those injected with CORT. Scale bar, 100 μm. Data are presented as mean ± SEM and were analyzed by two-way ANOVA with Bonferroni’s multiple comparison test. * Denotes differences between vehicle Wt and LCN2-null mice, and ^δ^ between vehicle and CORT Wt. *^,δ^
*p* ≤ 0.05, ^δδ^
*p* ≤ 0.01.

## Data Availability

The data presented in this study are available within the article or Appendix A.

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
