# Peer review of "The Effects of Stress on Hippocampal Neurogenesis and Behavior in the Absence of Lipocalin-2"

_ijms, 2023, doi:10.3390/ijms242115537_

Round 1
Reviewer 1 Report
The authors make a compelling case for their working hypothesis, well exposed and adequately backed by their data. Both the target protein and the mechanisms studied fall well within the scope of the journal, the section and the special issue this paper has been submitted to.
There are just a couple of minor details that I think could be considered:
The number of injections, either subcutaneous or intraperitoneal, administered to the experimental subjects was 38 each by my calculations, in a period of 33 consecutive days. That seems, to put it mildly, a lot. The proper institutional review board statement is included, as expected, so it is obvious that this treatment was approved by the relevant authorities and scientifically sound. However, I think it would be beneficial for the paper and my peace of mind for the authors to discuss a little bit more why was this approach selected instead of, say, the chronic stress induction they mention or implanted osmotic pumps to replace one or the two sets of injections.
Also, I'd like a clarification on the breeding protocol described in section 4.1. The crossing of heterozygous mice would produce a Mendelian 1:2:1 ratio of mice, with two different substrains of LCN2-null mice, carrying one or two copies of the mutation, respectively. Does this difference not affect the phenotype of the mice? Could the authors elaborate on which mice were used and why?
And it would be interesting to explain the rationale behind using only male mice, and specifically for that age. I can totally relate to the huge economic cost of maintaning a sizable experimental colony, but a short paragraph explaining whether or not gender or age differences have been described in this mutant could be useful.
Finally, there is an Y-axis cut in Figure 2.a. that I guess it's been added to emphasize the differences in Ki67+ cells, but I don't really think it is necessary and can be a little distracting.
Author Response
Reviewer: 1
Comments to the Author
The authors make a compelling case for their working hypothesis, well exposed and adequately backed by their data. Both the target protein and the mechanisms studied fall well within the scope of the journal, the section and the special issue this paper has been submitted to.
We acknowledge the reviewer remarks, comments and suggestions that greatly contribute to improve the manuscript.
There are just a couple of minor details that I think could be considered:
The number of injections, either subcutaneous or intraperitoneal, administered to the experimental subjects was 38 each by my calculations, in a period of 33 consecutive days. That seems, to put it mildly, a lot. The proper institutional review board statement is included, as expected, so it is obvious that this treatment was approved by the relevant authorities and scientifically sound. However, I think it would be beneficial for the paper and my peace of mind for the authors to discuss a little bit more why was this approach selected instead of, say, the chronic stress induction they mention or implanted osmotic pumps to replace one or the two sets of injections.
We understand the reviewer comment and, in the revised manuscript, we highlighted that limitation of this study. In the revised manuscript we added in line 204 “Although well described, and as we confirm, this is an effective model to induce stress response at behavioral and cellular level. However, other models should be considered, such as the usage of the chronic unpredictable stress protocol, or even the osmotic pumps to replace the injections that these animals received”.
Also, I'd like a clarification on the breeding protocol described in section 4.1. The crossing of heterozygous mice would produce a Mendelian 1:2:1 ratio of mice, with two different substrains of LCN2-null mice, carrying one or two copies of the mutation, respectively. Does this difference not affect the phenotype of the mice? Could the authors elaborate on which mice were used and why?
In a previous study (not published), we verified the phenotypes of the mice carrying one copy of the mutation, and observed no major significant changes in terms of behavior and cellular phenotypes, when compared to Wt littermate controls. In this manuscript, we analyzed the impact of the corticosterone treatment on neurogenesis in animals carrying two copies of the mutation, referred as LCN2-null mice, building on our previous published work regarding the role of LCN2 in neurogenesis and cognitive behavior (Ferreira AC et al 2018, Mol Psychiatry; PMID: 28485407 DOI: 10.1038/mp.2017.95).
And it would be interesting to explain the rationale behind using only male mice, and specifically for that age. I can totally relate to the huge economic cost of maintaning a sizable experimental colony, but a short paragraph explaining whether or not gender or age differences have been described in this mutant could be useful.
We acknowledge the reviewer comments and we now add this information in the materials and methods section of the revised manuscript. In line 242-247 we now add: “The usage of only males with 2 months of age is mainly related with the fact that, in previous studies, we have described that LCN2-null males with this age presented altered hippocampal neurogenesis and contextual discriminative behaviors [7]; however, it is not expected that this is gender- or age-specific, since the cognitive deficits and hippocampal neurogenesis defects were also detected in LCN2-null aged animals [33].”
Finally, there is an Y-axis cut in Figure 2.a. that I guess it's been added to emphasize the differences in Ki67+ cells, but I don't really think it is necessary and can be a little distracting.
A new figure 2.a. is now provided in the revised manuscript.
Reviewer 2 Report
The role of adolescence and adult neurogenesis in learning and memory is very intriguing and intensively studied in rodent models. However, in humans, it is quite a controversial topic. Lipocalin-2 (LCN2) modulates the neurogenesis. Authors used LCN2-null mice, which is also used for autoimmune studies, and glucocorticoid hormones (CORT) to modulate the neurogenesis in these mice negatively. The authors are trying to establish the role of stress in neurogenesis and learning and memory, which is interesting. However, I have a few concerns regarding the study.
Q-1 As the authors mentioned, chronic stress exposure negatively affects hippocampal neurogenesis. In the contextual discrimination experimental paradigm, mice received three light shocks at an interval of 2 min, and chambers were cleaned with 10% ethanol between each mouse. However, the contextual chamber needs to be cleaned with 70% ethanol to remove the traces of previous animals. Three shocks will give intense stress to the animal, which will affect the subsequent animal behavior because the author cleans the context with 10% ethanol.
Q-2 LCN2 is expressed in the hippocampus. olfactory bulb, cerebellum, etc., and LCN2 upregulate after stress. WT animals will express more LCN2 in the hippocampus as well as olfactory bulb. When the author cleans the context with 10% ethanol between the animals, it will be more stressful for WT, not for LCN2 null mice; that could be the reason the authors observed these changes.
Q-3 In the contextual discrimination experimental paradigm, authors need to counterbalance between the contexts A and B. Half of the animals must be exposed to context A first, then B, and vice versa.
Q-4 As the authors mentioned in their previous published paper (Ferreira et al., 2018), LCN2 deletion increased anxiety, depression, and memory impairment, as shown in Figures 3A and B. However, after CORT treatment to LCN2 null mice, Latency is trending. What's the author's explanation of that
Author Response
Reviewer: 2
The role of adolescence and adult neurogenesis in learning and memory is very intriguing and intensively studied in rodent models. However, in humans, it is quite a controversial topic. Lipocalin-2 (LCN2) modulates the neurogenesis. Authors used LCN2-null mice, which is also used for autoimmune studies, and glucocorticoid hormones (CORT) to modulate the neurogenesis in these mice negatively. The authors are trying to establish the role of stress in neurogenesis and learning and memory, which is interesting. However, I have a few concerns regarding the study.
We acknowledge the reviewer remarks, comments and suggestions that contribute to improve the manuscript.
Q-1 As the authors mentioned, chronic stress exposure negatively affects hippocampal neurogenesis. In the contextual discrimination experimental paradigm, mice received three light shocks at an interval of 2 min, and chambers were cleaned with 10% ethanol between each mouse. However, the contextual chamber needs to be cleaned with 70% ethanol to remove the traces of previous animals. Three shocks will give intense stress to the animal, which will affect the subsequent animal behavior because the author cleans the context with 10% ethanol.
We understand the point raised by the reviewer. This is a follow-up study of a previous work where we showed that, in basal conditions LCN2-null mice have deficits in the contextual discrimination experimental paradigm (Ferreira AC et al 2018, Mol Psychiatry; PMID: 28485407 DOI: 10.1038/mp.2017.95). In order to maintained the same protocol, and to better compare the results we followed the same protocol. The fact that we test the animals randomly and that all animals are submitted to the same procedure give us also. confidence in the results.
Q-2 LCN2 is expressed in the hippocampus. olfactory bulb, cerebellum, etc., and LCN2 upregulate after stress. WT animals will express more LCN2 in the hippocampus as well as olfactory bulb. When the author cleans the context with 10% ethanol between the animals, it will be more stressful for WT, not for LCN2 null mice; that could be the reason the authors observed these changes.
The point raised by the reviewer is relevant however, in basal conditions, we saw that the WT animals had a percentage of freezing that is around 80% which is, the normal freezing time for an WT animal as reported in the literature (Li et al., 2020, Elife 2020; PMID: 33215988 DOI: 10.7554/eLife.61882).
Q-3 In the contextual discrimination experimental paradigm, authors need to counterbalance between the contexts A and B. Half of the animals must be exposed to context A first, then B, and vice versa.
Herein we really understand the reviewer but again since this is a follow-up study, in order to maintain the same conditions, we followed the same protocol as published in (Ferreira AC et al 2018, Mol Psychiatry; PMID: 28485407 DOI: 10.1038/mp.2017.95).
Q-4 As the authors mentioned in their previous published paper (Ferreira et al., 2018), LCN2 deletion increased anxiety, depression, and memory impairment, as shown in Figures 3A and B. However, after CORT treatment to LCN2 null mice, Latency is trending. What's the author's explanation of that
We believe that because LCN2-null mice have basal high levels of corticosterone, as we reported in Ferreira et al 2013, these animals are not able to respond to the CORT treatment, as we observed with WT mice, probably due to a “ceiling effect”. We discussed this point in the present manuscript, lines 229-232.
Reviewer 3 Report
This research contributes to the understanding of the role of Lipocalin-2 in adult neurogenesis in rodents and the impact of stress on it.
Minor corrections are needed that could contribute to the comprehensibility of the work. As follows:
- in the first paragraph of the introduction, one gets the impression that all mammals have a significant capacity for adult neurogenesis, even though most of the aforementioned research was conducted on rodents. Clarify for which animal model the derived conclusions apply.
- the last sentence of the introduction is incomprehensible (external-dependent regulation?) cortisol given in a given dose is not the same as external-dependent regulation...
- use the full name instead of the abbreviation in the title 2.1
- state exactly how many animals were used for which of the experiments
- state how you arrived at the dose of 20 mg/kg of cortisol and the administration regimen
- state how you measured neuron survival
- with the explanation of the results by pictures, the explanation of the initial difference between WT and LCN2-null is missing. This initial difference complicates the understanding of the results because it would be more accurate if it were expressed as a % of cells and not as an absolute value (Fig 2)
- in the main text, some abbreviations are used without first being explained
- and in the quantification of behavioral tests, you used absolute values, although one could try to calculate the percentage of change here as well.
- in the discussion, you did not touch on the initial difference between groups of animals
- what is BrdU dissolved in before injection?
Author Response
Reviewer(s)' Comments to Author:
We appreciate the reviewer’s comments, that we will further address in detail.
Reviewer: 3
This research contributes to the understanding of the role of Lipocalin-2 in adult neurogenesis in rodents and the impact of stress on it.
We acknowledge the reviewer remarks, comments and suggestions that contribute to improve the manuscript.
Minor corrections are needed that could contribute to the comprehensibility of the work. As follows:
- in the first paragraph of the introduction, one gets the impression that all mammals have a significant capacity for adult neurogenesis, even though most of the aforementioned research was conducted on rodents. Clarify for which animal model the derived conclusions apply.
We have now specified in the revised manuscript that is mainly shown in rodents. Now in the revised manuscript in line 27 we specified: “In rodents, these newly generated functional neurons integrate the pre-existing neuronal circuitry, and contribute to local neural plasticity [1]. The generation of new neurons and their integration is crucial for hippocampal integrity and function, and it has been shown to directly modulate learning and memory, pattern separation, emotion, and even neurodegeneration [2-5].”.
- the last sentence of the introduction is incomprehensible (external-dependent regulation?) cortisol given in a given dose is not the same as external-dependent regulation...
We agree with the reviewer and we considered that the last sentence does not add relevant information to the message of the manuscript and at the same time it distorts the message. For that reason, we removed the sentence from the revised manuscript.
- use the full name instead of the abbreviation in the title 2.1
We really acknowledge this correction. Indeed title 2.1 was a mistake and a new title is now provided.
- state exactly how many animals were used for which of the experiments
The number of animals used in which experiments is described in each figure legend. However, to make it clear we add this information also in the materials and methods section. Line 254 we add that: “Regarding the number of animals used in this study, for the behavioral analysis 6-10 mice per group were used and for the assessments of cell proliferation and neural stem cells self-renewal and survival 4-6 mice per group were used.”,
- state how you arrived at the dose of 20 mg/kg of cortisol and the administration regimen
The dosage and the regiment of the injections were performed accordingly with the published literature, as it is stated in the materials and methods section (reference 16 and 17). However, we reinforced that information and make it clear in the revised manuscript. In the discussion section line 196 we now add: “The dosage and the regiment of the CORT injections were performed as previously reported [17] and, in fact, we confirmed the efficacy of the stress protocol here applied since both Wt and LCN2-null mice presented significant alterations in body, adrenals and thymus weight.”
- state how you measured neuron survival
Neuronal survival was measured by quantifying the total number of Calbindin/BrdU-double positive cells in the dentate gryus. For that, BrdU was i.p. injected for 5 consecutive days at the beginning of the treatment protocol, and number of cells generated was quantified 28 days later. We now clearly state that in the materials and methods section line 265-268: “As described above, experimental groups were intraperitoneally injected with 50 mg/Kg of BrdU (Sigma Aldrich), previously dissolved in saline solution, twice a day for 5 consecutive days at the beginning of the CORT treatment to assess neuron survival and maturation.”.
- with the explanation of the results by pictures, the explanation of the initial difference between WT and LCN2-null is missing. This initial difference complicates the understanding of the results because it would be more accurate if it were expressed as a % of cells and not as an absolute value (Fig 2)
We understand the reviewer point of view however, when we express as % of cells, for the WT and LCN2-null mice, that way we are considering that they have the same initial number of proliferating cells, which is not the case. In that sense we think that is better to represent the data as absolute values.
- in the main text, some abbreviations are used without first being explained
We have now revised the manuscript in accordance and we acknowledge the reviewer comment.
- and in the quantification of behavioral tests, you used absolute values, although one could try to calculate the percentage of change here as well.
Herein, when analyzing the behavioral, and express it as % for the WT and LCN2-null mice we are also considering that they have the same performance, which is not the case. In that sense we believe that is better to represent the data as absolute values.
- in the discussion, you did not touch on the initial difference between groups of animals
In order to address this and to make it clear, we now rephrase the sentence in the discussion in line 229: “As so, we cannot exclude the contribution of the sustained production of CORT in the absence of LCN2 in boosting hippocampal neurogenesis deficits and in promoting behavioral deficits that these animals presented already in basal conditions [7].”.
- what is BrdU dissolved in before injection?
We acknowledge this comment and we now add more precise information regarding the BrdU injections in line 265: “As described above, experimental groups were intraperitoneally injected with 50 mg/Kg of BrdU (Sigma Aldrich), previously dissolved in saline solution, twice a day for 5 consecutive days at the beginning of the CORT treatment to assess neuron survival and maturation.”.
Round 2
Reviewer 2 Report
Author's answered all my questions.